# Description of Ultra-Processed Food Intake in a Swiss Population-Based Sample of Adults Aged 18 to 75 Years

**DOI:** 10.3390/nu14214486

**Published:** 2022-10-25

**Authors:** Valeria A. Bertoni Maluf, Sophie Bucher Della Torre, Corinne Jotterand Chaparro, Fabiën N. Belle, Saman Khalatbari-Soltani, Maaike Kruseman, Pedro Marques-Vidal, Angeline Chatelan

**Affiliations:** 1Department of Nutrition and Dietetics, Geneva School of Health Sciences, HES-SO University of Applied Sciences and Arts Western Switzerland, Rue des Caroubiers 25, 1227 Carouge, Switzerland; 2Institute of Social and Preventive Medicine (ISPM), University of Bern, Mittelstrasse 43, 3012 Bern, Switzerland; 3Center for Primary Care and Public Health (Unisanté), University of Lausanne, Route de la Corniche 10, 1010 Lausanne, Switzerland; 4The University of Sydney School of Public Health, Faculty of Medicine and Health, University of Sydney, Sydney, NSW 2006, Australia; 5ARC Centre of Excellence in Population Aging Research (CEPAR), University of Sydney, Sydney, NSW 2006, Australia; 6MK-Nutrition, Rue Grosselin 25, 1227 Carouge, Switzerland; 7Department of Medicine, Internal Medicine, Lausanne University Hospital, Rue de Bugnon 21, 1011 Lausanne, Switzerland

**Keywords:** food processing, ultra-processed, NOVA classification, food group, macronutrients, Switzerland, Swiss adults, menuCH

## Abstract

Ultra-processed foods (UPFs) are associated with lower diet quality and several non-communicable diseases. Their consumption varies between countries/regions of the world. We aimed to describe the consumption of UPFs in adults aged 18–75 years living in Switzerland. We analysed data from the national food consumption survey conducted among 2085 participants aged 18 to 75 years. Foods and beverages resulting from two 24-h recalls were classified as UPFs or non-UPFs according to the NOVA classification, categorized into 18 food groups, and linked to the Swiss Food Composition Database. Overall, the median energy intake [P25–P75] from UPFs was 587 kcal/day [364–885] or 28.7% [19.9–38.9] of the total energy intake (TEI). The median intake of UPFs relative to TEI was higher among young participants (<30 years, *p* = 0.001) and those living in the German-speaking part of Switzerland (*p* = 0.002). The food groups providing the most ultra-processed calories were confectionary, cakes & biscuits (39.5% of total UPF kcal); meat, fish & eggs (14.9%); cereal products, legumes & potatoes (12.5%), and juices & soft drinks (8.0%). UPFs provided a large proportion of sugars (39.3% of total sugar intake), saturated fatty acids (32.8%), and total fats (31.8%) while providing less than 20% of dietary fibre. Consumption of UPFs accounted for nearly a third of the total calories consumed in Switzerland. Public health strategies to reduce UPF consumption should target sugary foods/beverages and processed meat.

## 1. Introduction

Ultra-processed foods (UPFs) are defined as “formulations of ingredients that result from a series of industrial processes (hence ‘ultra-processed’), many requiring sophisticated equipment and technology” [1]. UPFs include soft drinks, energy drinks, ready-to-eat salty snacks, chocolate, confectionery, ice cream, mass-produced packaged breads, margarines, pre-packaged biscuits, breakfast cereals, pre-prepared pies, pasta and pizza dishes, poultry and fish nuggets and sticks, sausages, burgers, hot dogs and other reconstituted meat products, industrial soups and sauces, and many other products [1]. In addition to added salt, sugars, oils, and fats, these industrial formulations include substances not used in homemade food preparations like colours, flavours, emulsifiers, and other additives, which are known as ultra-processing markers [1]. The NOVA classification designates four categories according to the extent of food processing: (group 1) unprocessed or minimally processed foods; (group 2) processed culinary ingredients; (group 3) processed foods; and (group 4) ultra-processed food and drink products (1). NOVA has been used to study the consumption of UPFs in different countries and regions of the world, their nutritional quality, and their association with various non-communicable diseases. These studies have shown that UPFs have unbalanced nutrient profiles, with high contribution of energy, saturated fatty acids (SFAs), added sugars, and sodium and low contribution of proteins, fibre, and most micronutrients [2,3,4]. In addition, their food matrix is modified so that the complex physical and nutritional structures of whole foods are lost during the food ultra-processing [5,6]. High consumption of UPFs has been associated with overweight/obesity [7,8,9,10,11], high waist circumference, metabolic syndrome, reduced high-density lipoprotein (HDL) cholesterol [7], as well as an increased risk of cardiovascular disease, cerebrovascular disease [7,8], cancers [8], and death [7]. 

The level of UPF consumption was reviewed in 21 countries with widely varying results [12], including a total of 1,378,454 subjects living in America, Europe, Asia, and Australia (no study in Switzerland). The United States (US) and the United Kingdom (UK) had the highest levels of consumption, reaching more than 50% of total energy intake (TEI); conversely, Italy had the lowest consumption (10–11%) [12]. Because Switzerland is a multilingual country (speaking mainly German, French, and Italian) and surrounded by three countries with differing dietary habits (Germany, France, and Italy), language-regional differences in UPF consumption are expected [13]. Furthermore, associations between consumption of UPFs and sociodemographic characteristics (e.g., sex, age, educational level, household income) as well as weight status have been found in several countries [14,15,16]. Considering sex, the levels of UPF intake appeared comparable, with men having often an overall slightly higher intake compared to women [12]. Regarding age, the highest levels of consumption were observed in children and adolescents and the lowest in older participants [12]. The association between education and consumption of UPFs is not consistent. In France, UPFs are consumed less by individuals with incomplete high school [15]. Conversely, in countries like Australia [17], Canada [18], and the US [14], the percentage of energy from UPFs was higher in lower educated participants. In Belgium, on the other hand, there were no differences in the consumption of UPFs between different levels of education [19]. When investigating the level of consumption of UPFs according to BMI, it was found that generally, the UPF intake was slightly higher in people with higher BMI [12]. In Switzerland, UPF consumption has been associated with excess body weight in women but not in men [11], but there is no information regarding the differential intake of UPFs by sociodemographic characteristics nor the contribution of UPFs to total nutrient intake. 

Nutritional surveillance of population-level dietary intake according to the level of processing by food group is necessary for setting goals, orienting policies, and monitoring the changes in diet quality and diet-related chronic diseases. Similarly, knowing how much of healthy or unhealthy nutrients is provided by UPFs in a standard diet is important for tailoring specific recommendations. Finally, determining whether the consumption of UPFs varies by sociodemographic subgroups makes it possible to tackle health disparities. These data are currently lacking in Switzerland. Therefore, the aims of this analysis of the first Swiss national food consumption survey, menuCH, were to (i) describe the consumption of UPFs according to sociodemographic characteristics; (ii) determine food groups that provide the most ultra-processed energy, and (iii) define the percentage of nutrients provided by UPFs in the Swiss diet.

## 2. Materials and Methods

### 2.1. Study Design and Population

We analysed the data from the Swiss National Nutrition Survey (menuCH; https://menuch.iumsp.ch, accessed on 21 April 2020), a cross-sectional survey conducted among non-institutionalised residents aged 18–75 years old (N = 2085) [13]. The stratified random sample was provided by the Federal Statistical Office. The participants were representative of the seven main regions of Switzerland and lived in the cantons of Aargau, Basel–Land, Basel–Stadt, Bern, Lucerne, St. Gallen, and Zurich (German-speaking region); Geneva, Jura, Neuchatel, and Vaud (French-speaking region); and Ticino (Italian-speaking region). The survey was conducted between January 2014 and February 2015. Pregnant and breastfeeding women were included. Institutionalised people or those with insufficient mobility to access a study centre were excluded, as well as people with insufficient oral and written language skills. The study was registered in the trial registry (identification number: ISRCTN16778734). Detailed information on the menuCH study design can be found in these references [13,20,21].

### 2.2. Dietary Assessment in the Swiss National Nutrition Survey

Fifteen trained dieticians assessed dietary intake via two non-consecutive 24-hour recalls (24HDR), the first being conducted face-to-face and the second by phone 2–6 weeks later. 24HDR were spread over all weekdays and seasons. To conduct 24HDR, dieticians used the computer-directed interview program GloboDiet^®^ (GD, formerly EPIC-Soft^®^, version CH-2016.4.10, International Agency for Research on Cancer (IARC), Lyon, France). The procedure was standardized and followed 3 steps: (i) general information about the participant (e.g., special diet, special day); (ii) quick list of food consumption occasions and items; and (iii) detailed description and quantification of all the consumed foods and beverages, including cooking and preservation methods, brand name, and portion size [22,23]. A book containing photos of standardised portions and a set of 60 household utensils (e.g., glasses, cups, plates) was used to estimate the consumed quantities [24]. The FoodCASE tool (Premotec GmbH, Winterthur, Switzerland) linked all consumed foods with the best match item of the Swiss Food Composition Database (2015 version) [25]. We included in our analysis energy and 7 nutrients: proteins; total carbohydrates; sugars (including all the mono and disaccharides, e.g., glucose, fructose, lactose, saccharose); dietary fibre; total fats; SFAs; and sodium. Other nutrients were excluded because more than 5% of the reported foods had missing data for these nutrients (e.g., calcium, vitamin D).

### 2.3. Food Classification According to Processing

A registered dietician (VBM) coded each food item as belonging (1) or not (0) to group 4 of the NOVA classification (foods and drinks). For foods considered as recipes by the GD software (e.g., sandwiches, salads, pizzas, lasagne), we classified each underlying ingredient independently. Alcoholic beverages were also classified according to their degree of processing. As previously described [26,27], we used “food descriptors” and “brand name” to ensure more accurate classification. For instance, the words “fresh”, “raw”, and “homemade” were characteristic of foods classified as not ultra-processed. Conversely, we considered descriptors such as “with flavour”, “industrial”, “pre-fried”, and “with artificial sweetener” as markers of ultra-processing. The online database Open Food Facts [28] and the websites of Swiss supermarkets were used to check the ingredient list of products and to facilitate decision-making, when relevant. When the level of processing was unclear for a food/beverage, the dietitian referred to a senior dietician (AC). In the absence of clear evidence of ultra-processing markers, a conservative attitude was adopted to avoid an overestimation of UPF consumption. 

### 2.4. Food Grouping

The GD software contains 18 main food groups. For this study, we reclassified foods into slightly modified groups according to their nutritional characteristics when there were discrepancies between GD and the Swiss Food Pyramid [29]. We (i) gather legumes, tubers, and cereal products; (ii) gather fruits and vegetables; (iii) separate nuts and seeds from fruits; (iv) separate ice-creams and milk-based desserts from dairy products; (v) gather meat with fish and eggs; (vi) separate breakfast cereals from cereal products; (vii) put avocado and olives with nuts and seeds. After reclassification, our 18 food groups were: cereal products, legumes & potatoes; fruit & vegetables; dairy products; meat, fish & eggs; added fats; nuts & seeds; industrial dishes; soups & broth; juices & soft drinks; other non-alcoholic beverages; alcoholic beverages & substitutes; sugar, honey, jam, sweet sauces & syrups; ice-creams & milk-based desserts; breakfast cereals; confectionary, cakes & biscuits; salty snacks; seasoning, spices, yeast & herbs; and other foods. Appendix A provides examples of foods from each food group.

### 2.5. Sociodemographic Characteristics

The participants completed a 49-item questionnaire at home, which was checked for completeness by the dieticians at the first interview [13]. The linguistic region was defined according to the home address of participants. An open question assessed the nationality (up to two countries) and participants were classified as Swiss or non-Swiss (foreigners). The number of people in the household was categorized into four categories: one, two, three, and four or more people. Education was dichotomized into (i) primary/secondary education (from no compulsory school to high school or specialized professional or vocational school) and (ii) tertiary education (university and higher vocational training, at least 5–7 years after compulsory school). 

### 2.6. Statistical Analyses

Descriptive statistics were used. Daily nutrient intake per survey participant was calculated as the mean intake of the two 24HDR. If the second 24HDR was missing (N = 28, 1.3% of the sample), data from the first 24HDR were used.

Medians and 25th and 75th percentiles (P25–P75) of TEI and energy intake from UPFs were calculated for the whole sample and by subgroups of participants. Medians were preferred over means because of the skewed distribution. Two-sample Wilcoxon rank-sum (Mann–Whitney) tests and Kruskal–Wallis equality-of-populations rank tests were used to determine if there were significant differences in the consumption of UPFs between groups, i.e., sex, age, linguistic region of residency, nationality, household size, and education (bivariate analyses). We also used multiple quantile regressions to test whether the potential differences between groups were still observed after adjustment for all the other parameters and monthly net household income (4499 CHF; 4500–8999 CHF; ≥9000 CHF; no answer) (1.00 CHF = 1.05 USD = 1.04 EUR, values as of 14 September 2022) (multivariable analyses). 

To assess the energy from UPFs (in kcal/day) for each of the 18 groups, means ± SD were computed because some medians were 0 and therefore not very informative. Weight of UPFs (in grams/day) in the total diet and by food group was also considered to better take heavy foods (e.g., beverages) and low-calorie foods (e.g., foods with artificial sweeteners) into account and to test whether the contribution of the food groups changed while taking weight or energy (kcal) into account.

We also calculated the medians and P25–P75 intake for 7 nutrients to understand how much UPFs contribute to total nutrient intake and therefore the nutritional benefits (and potential risks) of reducing UPF consumption. For these calculations, alcoholic beverages were excluded, as they are not part of an ideal diet [30]. The relative nutrient intakes of UPFs compared to total nutrient intakes were based on median intakes.

All statistical analyses were performed using STATA software, version 15 (Stata Corporation, College Station, TX, USA). A *p*-value of <0.05 was considered statistically significant. 

## 3. Results

### 3.1. Characteristics of the Participants

The total sample was composed of 2085 participants (Table 1). A flowchart showing the causes of participants’ exclusion from analyses is presented in Appendix A. The most represented participants were women (54.6%), participants aged 50 to 64 years (mean age of 46.3 ± SD 15.8), living in the German-speaking region (65.2%), of Swiss nationality (84.0%), living in households of two people (39.6%), and with primary/secondary education (51.3%). Four questionnaires (0.2%) were not returned.

### 3.2. Consumption of UPFs according to Characteristics of Participants

Overall, median TEI among participants was 2089 kcal [P25–P75: 1665–2552] (women 1842 vs. men 2417 kcal) and UPFs represented 28.7% of TEI [P25–P75: 19.9–38.9]. Consumption of UPFs was significantly higher among people aged 18 to 29 years (34.8% of TEI) than in older groups (e.g., 26.3% in 65–75-year-olds; *p* = 0.001). Consumption of UPFs was also significantly higher in people living in the German-speaking region (29.6% vs. 28.0% in the Italian-speaking region and 27.2% in the French-speaking region; *p* = 0.002) and among Swiss nationals (29.2% vs. 26.1% for non-Swiss; *p* = 0.002). Associations were also found between UPF consumption (% of TEI) and sex (higher among women, *p* = 0.012), and education (higher among people with lower education, *p* = 0.06). However, no differences in UPF consumption were found according to household size (*p* > 0.05) (Table 1). Seven people did not consume any UPFs during the two recorded days. 

### 3.3. Distribution of Energy Intake (Kcal) from UPFs by Food Group

Table 2 shows the distribution of energy intake from UPFs by food group in the whole sample. In total, the mean ± SD intake of UPFs was 676 ± 440 kcal, representing 31.0% of the mean TEI (2184 kcal) (results slightly different from medians presented in Table 1). Food groups that were the main energy contributors (Columns 1 and 2) were cereal products, legumes & potatoes (564 kcal; 25.6% of TEI); meat, fish & eggs (272 kcal; 12.6% of TEI); and dairy products (269 kcal; 12.4% of TEI). 

Salty snacks; confectionary, cakes & biscuits; and other foods, including meat substitutes or added artificial sweeteners were predominantly constituted of UPFs (100.0%, 99.6%, and 94.1%, respectively, Columns 3 and 4). Among UPFs, most calories came from confectionary, cakes & biscuits (204 kcal, 29.5% of total daily intake from UPFs, Column 5); followed by meat, fish & eggs (105 kcal, 14.9%); and cereal products, legumes & potatoes (78 kcal, 12.5%). Together, other foods; ice-creams & milk-based desserts; alcoholic beverages & alcoholic drink substitutes; soups & broth; industrial dishes; and other non-alcoholic beverages accounted for less than 10% of daily UPFs calories. The last two groups (i.e., nuts & seeds; and fruit & vegetables) did not provide ultra-processed energy (Table 2, Column 5). 

### 3.4. Distribution of Weight of Total Diet (Grams) from UPFs by Food Group

On average, participants consumed 3443 g (SD: 981) of foods and beverages per day, 481 g (SD: 463) (14.2%) of which were from UPFs (see Appendix A). The major contributors to UPF intake were juices & soft drinks (210 g, 26.0%), confectionary, cakes & biscuits (50 g, 15.9%), and dairy products (48 g, 11.1%, Figure 1). 

### 3.5. Contribution of UPFs to Intake of Macro- and Micronutrients 

UPFs accounted for 39.3% of the total daily intake of sugars, 32.8% of SFAs, 31.8% of total fats, and 30.7% of total carbohydrates (Figure 2). UPFs accounted for less than 20% of total daily intake for dietary fibre (15.2%). Details on absolute intakes and proportions of missing nutrient values are presented in Appendix A.

## 4. Discussion

### 4.1. Principal Findings

UPFs represent a substantial percentage of TEI (29%). We found a higher percentage of energy from UPFs among younger adults, those living in the German-speaking region, and Swiss nationals. Conversely, people aged 50–64 and 65–75 years and non-Swiss nationals were participants who consumed the least UPFs. Major contributors of ultra-processed calories were confectionary, cakes & biscuits; meat, fish & eggs; and cereal products, legumes & potatoes. These three food groups contributed to more than 50% of the energy intake from UPFs. When taking the weight of UPFs in the diet into account, food groups consumed in higher amounts were juices & soft drinks; and confectionary, cakes & biscuits. UPFs provided a large proportion of sugars, SFAs, and total fats. Conversely, the contribution of UPFs was below 20% for dietary fibre.

### 4.2. Consumption of UPFs according to Countries

A systematic review including several countries showed that the consumption of UPFs greatly varies between Western high-income countries, with the US and UK being the countries with the highest percent of TEI from UPFs (higher than 50%), and Italy being the country with the lowest level (about 10%) [12]. For instance, in Canada, the levels of intake were also elevated (more than 45%). Australia showed levels of UPF consumption ranging from 38.9% to 42.0% of TEI. In Europe, in both Spain and France the consumption varied between 17.0% and more than 30%, depending on the studies. Consumption in Belgium was similar to consumption in Switzerland (means of 30.3% and 31.0%, respectively), while in Portugal the intake was lower (22.2%) but higher than in Italy [12].

### 4.3. Consumption of UPFs according to Characteristics of Participants

We found that the highest percentage of energy intake from UPFs was in young adults (<30 years) and decreased with age. This trend has already been observed in previous studies [15,16,17]. Young adults might be attracted by the convenience (limited time spent in the kitchen) of these products [31]. Interestingly, when we related the time required to cook a hot meal at home during a usual week with the consumption of UPFs in menuCH participants, we found that those who spend less than 30 min cooking had a significantly higher percentage of kilocalories from UPFs (Appendix A). Other authors also showed that time spent on food preparation at home was associated with indicators of diet quality and frequency of fast-food restaurant use [32]. In addition, among adolescents and young adults, the use of social media is high, and greatly promotes the consumption of branded UPFs, such as soft drinks, cakes, crisps, pizzas, and sweets [33]. 

People from the German-speaking region consumed more UPFs. This is consistent with previous literature showing that people from the German-speaking region less frequently cook hot lunches themselves at home in comparison to people from the French-speaking and Italian-speaking regions [34]. Furthermore, the consumption of UPFs, such as soft drinks (including fruit lemonades and sugar-free soft drinks) or processed meat is higher in the German-speaking part of Switzerland [13]. 

In the current study, non-Swiss nationals consumed significantly fewer UPFs, even though this group was slightly underrepresented in the sample [13]. The majority of foreigners residing in Switzerland are Italian, German, Portuguese, and French nationals [35]. People from Italy, Portugal, and France may have maintained a diet closer to the Mediterranean diet, which is usually poor in UPFs [36]. Indeed, when the adherence to the Mediterranean diet over 50 years was analysed in 41 countries, Germany ranked 35th and Switzerland 34th, while Portugal, Italy, and France ranked 10th, 14th, and 27th, respectively [37]. Moreover, another study showed that the average household availability of UPFs was lower in Portugal, Italy, and France compared to other European countries such as Germany or Austria (Switzerland not included in this analysis) [38]. Of note, the same phenomenon was found in Australia and Canada, where the intake of UPFs was also significantly lower among immigrants compared to locals [16,17].

Energy intake from UPFs only slightly differed according to education. Other barriers than lower education like taste, daily habits, and lack of time and willpower may play a role in adherence to healthy eating [39]. Furthermore, in this study, the intake from minimally or unprocessed foods was not investigated. It is possible that, even if the consumption of UPFs was similar, foods of NOVA group 1 were more consumed by people with higher education, as demonstrated in Belgium by Vandevijvere et al. [19]. This could be explained by the fact that people with higher education are more health conscious [40,41,42]. 

### 4.4. Distribution of Energy Intake from UPFs by Food Group

Ultra-processed energy came mainly from confectionary, cakes & biscuits; meat, fish & eggs; and cereal products, legumes & potatoes. Comparing our results with other studies is difficult because the way foods are grouped differs from one study to another. However, a study conducted in 22 European countries reported that the two main UPFs consumed among adults were fine bakery wares and sausages [43]. In our study, chocolate, industrial cakes, and cookies are typical UPFs of the group confectionary, cakes & biscuits. Because Swiss people consume the most chocolate per capita worldwide [44], this could explain why confectionary, cakes & biscuits was the food group contributing most to ultra-processed energy.

### 4.5. Distribution of Intake from UPFs (Grams/Day) by Food Group

The average consumption of UPFs in adults across 22 European countries was estimated at 328 g/day, representing an average share of total weight intake of 12% [43]. In our study, these figures were slightly higher: 481 g/day and 14.2%, respectively. A possible explanation is that alcoholic beverages were not considered in the international study. When the proportion (in weight, g/day) of UPFs in the total diet was analysed, major contributors were juices & soft drinks; confectionary, cakes & biscuits; and dairy products. Across Europe, the most consumed ultra-processed drinks were soft drinks and fruit/vegetable juices [43]. This analysis shows that the UPFs preferred by consumers are similar in Switzerland. 

### 4.6. Nutrition Profile of UPFs 

We found that diets rich in UPFs are high in sugars and fats, especially SFAs, and low in fibre, which is in line with other studies [18,45,46]. In this study, UPFs contributed nearly 40% of total sugar and 35% of SFA intake—nutrients that have been associated with a greater risk of chronic diseases [47]. The contribution of sodium was almost 30%, and it is known that a reduction in sodium intake reduces blood pressure [48,49]. In the US diet, the average intake of carbohydrates, added sugars, and SFAs increased significantly with the dietary contribution of UPFs [2]. In the UK, UPFs contributed nearly 65% of all free sugars (different from total sugars) in all age groups [50], and the intake of carbohydrates, free sugars, total fats, SFAs, and sodium increased significantly as UPF consumption increased [51]. In France, UPFs represented most of the total and free sugars and total fats, SFAs, but only a minor part of proteins and fibre [15]. Because of the poor nutritional profile of UPFs, high intake affects people’s health, and the risk of several non-communicable diseases is higher [52,53,54]. Thus, replacing UPFs with less- or un-processed foods could improve the quality of the diet without drastically impacting the intake of proteins [55]. Of note, in our study, values in unsaturated fatty acids and micronutrients were more likely to be missing from the Food Composition Database for UPFs than for non-UPFs, which limited the analysis for these nutrients(Appendix A).

### 4.7. Strengths and Limitations

For the assessment of dietary intake we used two 24HDRs, which may have led to misreporting of intake due to social desirability and recall bias [56]. However, 24HDRs are appropriate for estimating average levels of food consumption in nutrition population-based surveys [56] and to describe UPF consumption in a given population [57]. Although we assessed diet in the whole of Switzerland, the number of participants from the Italian-speaking region, a small region in Switzerland, was limited in our sample. The categorization of groups does not always make it possible to distinguish foods within the 18 food groups that are ultra-processed, although Appendix A provides specific examples of ultra-processed products in each group. In addition, food description did not always contain enough information to categorize foods according to the NOVA classifications with certainty; our conservative approach might have underestimated UPF consumption. Finally, micronutrient content was not available for all foods/beverages, thus limiting the number of nutrients included in our analysis. 

Despite these limitations, this is the first study to assess the importance of UPFs in a representative sample of the Swiss population encompassing three linguistic regions. The inclusion of two non-consecutive 24HDR conducted by trained dieticians enabled the estimation of detailed dietary intake (e.g., systematic description of cooking and preservation methods, brand names, etc.), allowing accurate identification of NOVA group 4 foods/beverages. Furthermore, the classification of foods (UPFs vs. non-UPFs) was performed by trained dieticians and discussed in case of discrepancies. 

## 5. Conclusions

Consumption of UPFs accounts for nearly one-third of total calories consumed in Switzerland, and their nutritional profile is unbalanced. Non-communicable disease prevention programs should especially target young adults. Nutritional education messages for reducing UPF consumption should first focus on the highest-contributing food groups, i.e., sugary foods/beverages and processed meat. Additionally, population-based public health measures, such as (i) taxing soft drinks or other UPFs, (ii) front-of-pack warning labels on NOVA 4 products, and (iii) school food policies banning UPFs from school meals, are possible strategies to reduce UPF consumption and prevent non-communicable diseases [58].

## Figures and Tables

**Figure 1 nutrients-14-04486-f001:**
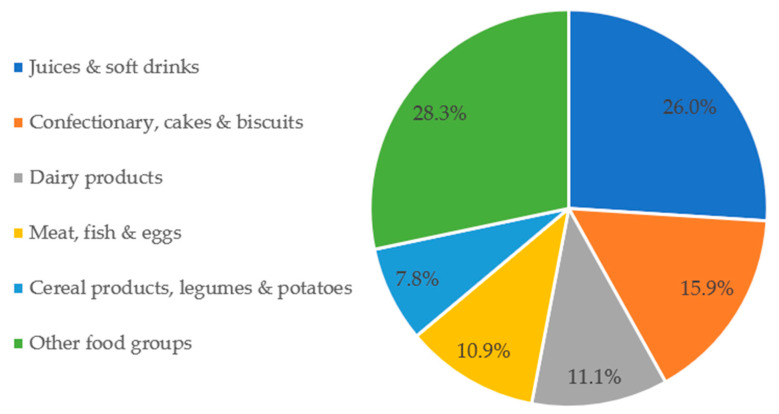
Proportion of UPF intake weight (grams/day) in comparison to the total diet weight, by major food group contributors. Seven people did not consume any UPFs (N_total_ = 2078).

**Figure 2 nutrients-14-04486-f002:**
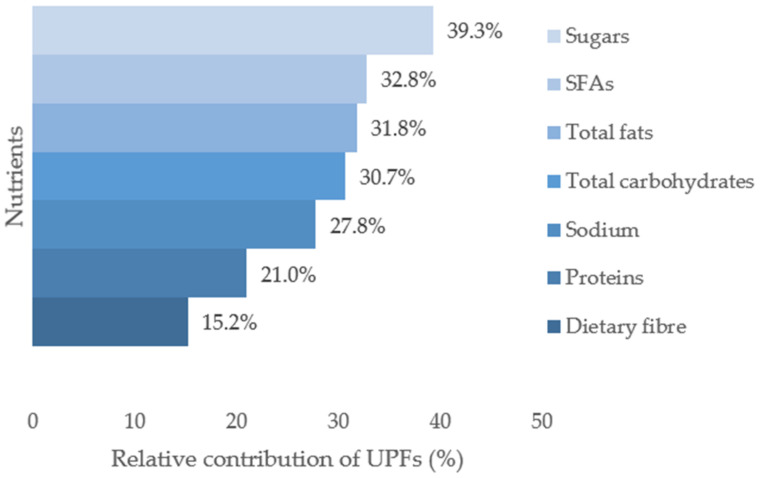
Relative contribution of UPFs to total daily intake (% based on medians) for seven nutrients. Sugars include all mono and disaccharides, e.g., glucose, fructose, lactose, saccharose; SFAs: saturated fatty acids.

**Table 1 nutrients-14-04486-t001:** Consumption of ultra-processed foods by sociodemographic characteristics. Swiss population aged 18 to 75 years, National Nutrition Survey 2014–2015.

Characteristics	N (%)	TEI (kcal/Day)		UPF Consumption (kcal/Day) ^1^		UPF Consumption (%TEI) ^2^		*p*-Value ^3^	*p*-Value ^4^
		Medians	P25–P75	Medians	P25–P75	Medians	P25–P75		
All participants	2085 (100.0)	2089	1665–2552	587	364–885	28.7	19.9–38.9		
Sex								0.125	0.012 *
Women	1139 (54.6)	1842	1527–2216	517	325–746	28.4	19.4–38.5		
Men	946 (45.4)	2417	1987–2993	703	445–1056	29.2	20.8–39.9		
Age groups, years								0.001 *	0.001 *
18–29	407 (19.5)	2221	1709–2731	727	478–1060	34.8	24.5–45.0		
30–39	327 (15.7)	2126	1700–2669	646	418–963	31.8	22.3–42.0		
40–49	450 (21.6)	2110	1702–2583	599	380–883	28.2	20.3–37.8		
50–64	562 (27.0)	2021	1640–2507	519	308–811	25.5	16.9–36.6		
65–75	339 (16.3)	1978	1641–2331	495	314–714	26.3	17.1–35.0		
Linguistic region								0.003 *	0.002 *
German	1359 (65.2)	2153	1721–2612	617	399–915	29.6	20.9–39.6		
French	510 (24.5)	1991	1647–2467	526	323–789	27.2	17.7–37.1		
Italian	216 (10.4)	1930	1515–2319	509	298–820	28.0	16.9–39.4		
1st nationality ^5^								0.009 *	0.002 *
Swiss	1751 (84.0)	2078	1665–2550	595	379–894	29.2	20.3–39.0		
Non-Swiss	330 (15.8)	2124	1654–2571	557	318–839	26.1	17.5–37.1		
Household size ^5^								0.060	0.400
One person	338 (16.2)	1996	1621–2446	573	330–892	29.0	18.5–40.6		
Two people	825 (39.6)	2070	1669–2514	565	353–835	28.1	19.7–37.3		
Three people	336 (16.1)	2103	1728–2522	591	371–901	28.8	19.5–39.7		
Four people and more	582 (27.9)	2132	1688–2678	626	407–945	30.2	21.5–40.1		
Education ^5^								0.073	0.060
Primary & secondary	1069 (51.3)	1993	1588–2495	574	355–894	29.1	20.2–39.7		
Tertiary	1012 (48.5)	2160	1762–2617	604	373–870	28.4	19.6–38.4		

^1^ Total energy intake from UPFs. ^2^ Contribution of UPFs from total energy intake. ^3^ Differences in UPF consumption as the percentage of total energy intake were tested with two-sample Wilcoxon rank-sum (Mann–Whitney) tests for sex and nationality. Kruskal–Wallis equality-of-populations rank tests were used for age, linguistic region, and household size. ^4^ Differences in UPF consumption as the percentage of total energy intake were tested using multiple quantile regressions. ^5^ Four questionnaires were not completed (N_total_ = 2081). * *p* < 0.05. TEI: total energy intake. UPFs: ultra-processed food and drink products. P25–P75: 25th and 75th percentiles. CHF: Swiss franc.

**Table 2 nutrients-14-04486-t002:** Distribution of energy intake (kcal) from UPFs by food group, in decreasing order (N = 2085, **bold** = 3 largest numbers, *italic* = 3 smallest numbers, by column).

Food Groups	Total Intake (kcal/Day)	Contribution to TEI (%TEI)	UPF intake (kcal/Day)	UPF intake from Total Intake (%) ^1^	UPF Intake from TEI (%TEI) ^2^
	Mean (SD)	Mean (SD)	Mean (SD)	Mean (SD)	Mean (SD)
Confectionary, cakes & biscuits	204 (216)	9.0 (8.3)	**204 (214)**	**99.6 (4.6)**	**29.5 (23.9)**
Meat, fish & eggs	**272 (218)**	**12.6 (8.8)**	**105 (150)**	35.3 (34.1)	**14.9 (18.6)**
Cereal products, legumes & potatoes	**564 (310)**	**25.6 (10.4)**	**78 (109)**	14.8 (19.3)	**12.5 (16.8)**
Juices & soft drinks	97 (150)	4.1 (5.5)	65 (136)	53.9 (44.4)	8.0 (13.3)
Dairy products	**269 (208)**	**12.4 (8.3)**	50 (86)	16.6 (25.0)	7.9 (13.9)
Seasoning, spices, yeast & herbs	95 (100)	4.4 (4.2)	33 (62)	32.2 (37.5)	5.5 (10.1)
Added fats	182 (152)	8.3 (6.0)	30 (75)	15.8 (25.6)	4.9 (10.0)
Salty snacks	22 (75)	1.0 (2.8)	22 (75)	**100.0 (0.0)**	3.0 (8.5)
Sugar, honey, jam, sweet sauces & syrups	60 (75)	2.7 (3.2)	18 (46)	26.4 (38.0)	2.9 (7.0)
Breakfast cereals	29 (72)	1.2 (3.1)	19 (58)	65.9 (44.9)	2.7 (8.1)
Other foods	*14 (58)*	*0. 7 (2.7)*	13 (58)	**94.1 (23.0)**	2.4 (9.3)
Ice-creams & milk-based desserts	22 (54)	1.0 (2.3)	14 (38)	74.4 (42.0)	2.3 (6.5)
Alcoholic beverages substitutes	107 (159)	4.7 (6.5)	13 (47)	14.4 (29.6)	1.9 (6.4)
Soups & broth	21 (55)	1.0 (2.8)	5 (29)	40.5 (48.3)	0.7 (4.3)
Industrial dishes	*13 (65)*	*0.6 (2.6)*	6 (45)	40.2 (48.3)	0.6 (4.4)
Other non-alcoholic beverages	*15 (33)*	*0.8 (1.5)*	*2 (12)*	*2.26 (13.3)*	*0.2 (2.3)*
Nuts & seeds	39 (84)	1.7 (3.6)	*0 (0)*	*0.0 (0.0)*	*0.0 (0.0)*
Fruit & vegetables	159 (120)	7.8 (6.1)	*0 (0)*	*0.0 (0.0)*	*0.0 (0.0)*
**Total**	2184 (750)	100.0	676.3 (440.1)	-	100.0

^1^ Among consumers only (N varies according to food groups, e.g., N = 2074 for cereal products, legumes & potatoes to N = 155 for industrial dishes). ^2^ Seven people did not consume any UPFs (N_total_ = 2078).

## Data Availability

The whole dataset and relevant documents (e.g., questionnaires, GloboDiet data) are accessible in the repository: https://menuch.iumsp.ch (accessed on 21 October 2022).

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
