# Peer review of "Description of Ultra-Processed Food Intake in a Swiss Population-Based Sample of Adults Aged 18 to 75 Years"

_nutrients, 2022, doi:10.3390/nu14214486_

Round 1
Reviewer 1 Report
It is an interesting article that brings relevant information to the literature. However, it needs structural changes to better understand and compare it with the literature.
Lines 67-72: The association between income and consumption of ultra-processed foods is not consistent. In some countries such as Brazil and Korea, ultra-processed foods are more consumed by higher income classes. In countries like Australia and the USA, consumption of ultra-processed foods is higher among lower-income individuals. In Canada and France, on the other hand, there is no difference in consumption between income classes. Correct sentence.
Line 165: Is the estimated income per household or per capita? If it is per household, it is necessary to correct for per capita income. From what I understand, you have a variable with the number of residents per household, right? If it is not possible to estimate per capita income, I recommend excluding the income variable from the analysis.
Line 184 (methods) and Table 2 (results): The categorization of groups is quite confusing. First, I strongly recommend that the definition of the subgroups follow the same line as other articles published using the NOVA classification. In the meat group, for example, there are fresh meats, processed meats and ultra-processed meats. In the sending manuscript we cannot distinguish processed meats from minimally processed meats.
We know that disaggregating culinary preparations into minimally processed foods and in natura foods is a challenge, but it would at least be necessary to distinguish processed foods. Processed foods are very important culturally and represent an important part of the calories consumed in European countries.
Figure 1: The figure can lead the reader to misinterpretations. As ultra-processed foods make up only 28% of calories consumed, it is to be expected that the contribution of this group to nutrient consumption is low. Again, I recommend reading scientific articles describing the consumption of ultra-processed foods in other countries and adopting similar methods. Ideally, I recommend presenting the attributable fraction of the nutrient intake according to all groups of the NOVA classification.
Line 285: Again, I would avoid statements about the price of ultra-processed foods, as there is no consistency in the literature on this.
Supplementary material: I could not access it.
Line 304: improve sentence, is a little confused.
Line 314-342: Again I reinforce the need to correct the analyses, using per capita income and the four groups of the NOVA classification. Otherwise the results are inconclusive.
Line 357: If the nutritional table is not complete, I recommend not evaluating nutrients with missing information. Otherwise the analyzes are underestimated and inconclusive.
Reviewer 2 Report
The article manuscript "Description of ultra-processed food intake in a Swiss population-based sample of adults aged 18 to 75 years" is well written. This paper focuses on a special food category called ultra-processed food. It involves the relationship between the description of the ultra-processed food and the customers’ many representative features, including gender, age, educational background, linguistic region, household size, and salary. This investment is valuable since the ultra-processed foods hazards are attracting public attention and the number of consumers is growing. Through this investment, we can figure out the customer profiles of this ultra-processed food in Switzerland. The government could also propose some direct policies or diet recommendations to help those people improve their health conditions based on the conclusion of this essay. The submission is worth publication. However, there are a few flaws that must be solved.
1. In the introduction section, the description of ultra-processed food is too ambiguous. Some specific foods should be listed. it is recommended that the author set up some model food of ultra-processed foods.
2. In Section 2.4 “food grouping”, the authors need to explain the reason for reclassifying the GD classification.
3. In Section 2.5 “Sociodemographic characteristics”, the community location should also be considered. The diet of people living in a big city and a village may be very different.
4. In the discussion section, the authors only choose the Europe countries as the compared example. Some American, Asian, or Australian countries should also be discussed to illustrate the condition of intaking ultra-processed food in Switzerland. As the author mentioned, the United States (US) and the United Kingdom (UK) had the highest levels of ultra-processed food consumption. So, it is necessary to include other countries in the discussion instead of just viewing European countries.
5. In Section 4.1 “Principal findings”, the authors only mentions the customers who consumed the most ultra-processed food. The people who consumed the least ultra-processed food should also be listed. To compare these two extreme groups’ features, the potential reason for the popularity of ultra-processed food may be revealed, which could directly guide the UPFs prevention programs’ policies.
6. In Section 4.5 “Nutrition profile of UPFs”, the relevance of the nutritional profile of the ultra-processed food and health condition is not sufficiently demonstrated. The author needs to provide more references that can prove intaking ultra-processed food does affect people’s health.
Reviewer 3 Report
Overconsumption of ultra-processed food remains serious potential of health risk. The present article described the ultra-processed food intake of adults aged 18 to 75 years in Switzerland. The results obtained would provide useful evidence for policy-making. Some minor issues were pending:
1. In section of Materials and Methods, a flowchart would be more understandable and instructive.
2. In sections of Discussion and/or Conclusion, further feasible recommendations that reduce the intake of ugary foods/beverages and processed meat would be more beneficial for public health nutrition policy-making.
